# Evaluation of Radiolabeled Girentuximab In Vitro and In Vivo

**DOI:** 10.3390/ph11040132

**Published:** 2018-11-28

**Authors:** Tais Basaco, Stefanie Pektor, Josue M. Bermudez, Niurka Meneses, Manfred Heller, José A. Galván, Kayluz F. Boligán, Stefan Schürch, Stephan von Gunten, Andreas Türler, Matthias Miederer

**Affiliations:** 1Department of Chemistry and Biochemistry, University of Bern, 3012 Bern, Switzerland; tais.basaco@dcb.unibe.ch (T.B.); josue.moreno@itg-garching.de (J.M.B.); niurka.meneses@dcb.unibe.ch (N.M.); stefan.schuerch@dcb.unibe.ch (S.S.); andreas.tuerler@dcb.unibe.ch (A.T.); 2Laboratory of Radiochemistry, Paul Scherrer Institute (PSI), 5232 Villigen PSI, Switzerland; 3Clinic for Nuclear Medicine, University Medical Center Mainz, 55131 Mainz, Germany; stefanie.pektor@unimedizin-mainz.de; 4Department for Biomedical Research (DBMR), University of Bern, 3010 Bern, Switzerland; manfred.heller@dbmr.unibe.ch; 5Institute of Pathology, University of Bern, 3010 Bern, Switzerland; jose.galvan@pathology.unibe.ch; 6Institute of Pharmacology (PKI), University of Bern, 3010 Bern, Switzerland; kayluz.frias@pki.unibe.ch (K.F.B.); stephan.vongunten@pki.unibe.ch (S.v.G.)

**Keywords:** carbonic anhydrase IX, girentuximab, renal cell carcinomas, ^177^Lu-radiopharmaceuticals, radioimmunotherapy

## Abstract

Girentuximab (cG250) targets carbonic anhydrase IX (CAIX), a protein which is expressed on the surface of most renal cancer cells (RCCs). cG250 labeled with ^177^Lu has been used in clinical trials for radioimmunotherapy (RIT) of RCCs. In this work, an extensive characterization of the immunoconjugates allowed optimization of the labeling conditions with ^177^Lu while maintaining immunoreactivity of cG250, which was then investigated in in vitro and in vivo experiments. cG250 was conjugated with S-2-(4-isothiocyanatobenzyl)-1,4,7,10-tetraazacyclododecane tetraacetic acid (DOTA(SCN)) by using incubation times between 30 and 90 min and characterized by mass spectrometry. Immunoconjugates with five to ten DOTA(SCN) molecules per cG250 molecule were obtained. Conjugates with ratios less than six DOTA(SCN)/cG250 had higher in vitro antigen affinity, both pre- and postlabeling with ^177^Lu. Radiochemical stability increased, in the presence of sodium ascorbate, which prevents radiolysis. The immunoreactivity of the radiolabeled cG250 tested by specific binding to SK-RC-52 cells decreased when the DOTA content per conjugate increased. The in vivo tumor uptake was < 10% ID/g and independent of the total amount of protein in the range between 5 and 100 µg cG250 per animal. Low tumor uptake was found to be due to significant necrotic areas and heterogeneous CAIX expression. In addition, low vascularity indicated relatively poor accessibility of the CAIX target.

## 1. Introduction

Targeted therapy with monoclonal antibodies (mAbs) carrying radioisotopes (radioimmunotherapy, RIT) has become a powerful tool in nuclear medicine, because of highly selective small molecules and targeted internal radiotherapy; it is currently being increasingly applied in the treatment of a growing number of malignant diseases [1,2,3,4,5,6,7,8]. Additionally, imaging studies with gamma emitting isotopes open the possibility of translational investigation of dosimetry [9]. The effectiveness of RIT depends on a number of factors related to the Ab such as the specificity and affinity as well as the immunoreactivity, stability and blood clearance of the resulting radioimmunoconjugate [10,11]. Three radiolabeled mAbs—two murine, ^90^Y-ibritumomab tiuxetan (Zevalin; Biogen Idec) and ^131^I-tositumomab (Bexxar; Corixa/GSK), and one chimeric, ^131^I-ch-TNT (Shanghai Medipharm Biotech)—have been approved for non-Hodgkin’s lymphoma (NHL) or lung cancer, but only a minority of anticancer mAbs currently in the clinics, are radioimmunoconjugates [12]. A suitable therapeutic window exists, only when RIT leads to a sufficient deposition of radioactivity in the tumor and, at the same time, acceptably low doses to healthy organs and tissues. Clinical studies revealed the limitations of radioimmunoconjugates as cancer therapeutics (for example, mAbs did not deliver effective radiation doses, especially to solid tumor sites [13], complex chemistry was often required for conjugation, and there were potentially toxic effects on normal tissues). In addition, in solid large tumors, accumulation of radioactivity by RIT is negatively influenced by slow diffusion properties of large molecules when interstitial fluid pressure is high and perfusion and vascular permeability are heterogeneous [10,14]. Furthermore, antigen expression can be reduced due to the presence of necrotic areas in those tumors, leading to a reduction of the radionuclide uptake.

Reaching cancer cells within solid tumors by mAbs, follows slower kinetics, compared to small molecules, such as peptides. Thus, long circulation times require therapeutic radionuclides with relatively long half-lives and with high stability of the radioimmune-conjugation. The released energy from the radiometal can also induce radiolysis and degradation of the protein, and thus loss of specificity. The formation of free radicals can be attenuated with the addition of quenching reagents like human serum albumin (HSA), gentisic acid, ascorbic acid, and other antiradicals [15]. β^−^ emitting radionuclides are currently prevalent in RIT since they have shown particular efficacy against a larger number of diseases. Radionuclides such as ^131^I, ^90^Y, ^188^Re, and ^177^Lu have been used extensively for the RIT of neoplastic lesions [3,16,17,18]. ^177^Lu is increasingly being used as a potent radionuclide for use in in vivo therapy because of its favorable decay characteristics. It decays by the emission of beta particles with maximum energies of 497 keV (78.6%), 384 keV (9.1%), and 176 keV (12.2%) to stable ^177^Hf [19] allowing delivery of therapeutic doses to the tumor and minimal doses to healthy tissues (i.e., low renal toxicity). The β-emission energy of ^177^Lu (β_mean_ = 166 keV) is lower than for other radionuclides commonly used for therapy such as ^131^I (β_mean_ = 191 keV), ^90^Y (β_mean_ = 699 keV), or ^188^Re (β_mean_ = 770 KeV). In addition, the emission of gamma rays of 113 keV (6.4%) and 208 keV (11%) with relatively low abundances, provides advantages that allow simultaneous scintigraphic studies, which help to monitor proper in vivo localization of the injected radiopharmaceutical and to perform dosimetric evaluations. Another important aspect for consideration is the relatively long t_1/2_ of ^177^Lu (6.73 days), which provides logistical advantages that facilitate its supply to locations far from reactors. The relatively long t_1/2_ also favors this nuclide for use in RIT [20]. 

An essential criterion for successful targeted radiotherapy with ^177^Lu depends on the choice of the proper bifunctional chelator (BFC). ^177^Lu is a bone-seeking element [14], its premature release and accumulation in the bone can lead to dose-limiting bone marrow toxicity. A number of acyclic and cyclic ligand systems have already been investigated for the labeling of mAb (full-length and fragments) with ^177^Lu [21,22,23]. Macrocyclic BFCs such as 1,4,7,10-tetraazacyclododecane-1,4,7,10-tetraacetic acid (DOTA) are of particular interest for the use of lanthanides in RIT because they are very well organized structures, enabling the formation of metal complexes, with a high thermodynamic stability and slow decomposition kinetics [22,24,25,26].

During the preparation of conjugated mAbs and the radiolabeling process, the protein might suffer some modifications, resulting in the loss of target recognition [27,28,29]. First, binding affinity to target and nontarget tissue might be affected during the immunoconjugation step, because the amino acids or peptides involved in the recognition of the antigen can be occupied by the BFC [11]. Amino acids with side chains amenable to modifications are found in all regions of Abs. Therefore, modification methods are not site-specific and there is no control over which amino acids are modified. First, immunoconjugates with modifications in the binding structures result in decreased efficacy of the targeting system [30,31,32]. Second, the sensitivity of Abs to heating and extreme pH values during the labeling process. The labeling reaction with DOTA and its analogues is very slow and largely depends on the conditions at which it is performed, which includes DOTA-mAb concentration, reaction temperature, time, pH, buffer used and its concentration, and the presence of metal ions such as Zn^2+^ and Fe^3+^. In order to achieve a good radiolabeling yield it is necessary to increase the temperature (>50 °C). The released energy from the radiometal might induce radiolysis and degradation of the protein, and thus loss of the specificity of the mAb to reach the target. The formed free radicals during the labeling can be attenuated with the addition of quenching reagents like HSA, gentisic acid, ascorbic acid, and others antiradicals [15].

Girentuximab (cG250) is a chimeric mAb reactive to the protein carbonic anhydrase IX (CAIX), a transmembrane glycoprotein with an intracellular enzymatic domain, which is overexpressed in hypoxic cells [33,34,35]. This antigen CAIX is also expressed on a variety of other solid tumors, including cervical, bladder, colon, and non-small cell lung cancer. Immunohistochemical (IHC) analysis of renal tumors showed homogeneous expression of CAIX in the vast majority (>80%) of primary renal cell carcinomas (RCCs) and about 70% of metastatic RCC lesions [36,37,38,39]. In most of the RCCs constitutive CAIX expression is common due to the presence of von Hippel Lindau mutation leading to a hypoxia inducible factor 1 alpha (HIF-1a), without expression in normal kidney tissue [34,36,38,39,40,41,42]. Therefore, CAIX is a promising target for renal cancer and other hypoxic tumors. A clinical trial failed to show a benefit of native cG250 in adjuvant treatment after surgery on the RCCs [43]. However, the patient collective showed a better overall prognosis than expected, which points to a lack of tumor cells that were available for targeting. Safety and tolerability of the antibody was demonstrated. Clinical and preclinical attempts have been made to use cG250 as carrier molecules for radioisotopes [44,45]. Therefore, there are many ongoing experiments in the therapeutic area. The first clinical trial for therapy was carried out with [^131^I]-girentuximab but the results were not as positive as expected. The widely used beta emitting isotope ^177^Lu (t_1/2_ = 6.7 days, 497 keV end-point energy) has also been investigated in preclinical and clinical studies treating metastasized RCCs [2,46,47,48].

Here, a detailed evaluation of the parameters affecting conjugation of S-2-(4-isothiocyanatobenzyl)-1,4,7,10-tetraazacyclododecane tetraacetic acid (p-SCN-Bn-DOTA) to girentuximab by the isothiocyanate group was conducted [49]. In order to characterize the conjugated cG250 the modification sites and the number of BFC modifications per mAb were determined by mass spectrometry. The loss of immunoreactivity post conjugation was evaluated in cells and tumor tissues. DOTA(SCN)-girentuximab conjugates with different BFC/mAb ratios were labeled with ^177^Lu and the influence of sodium ascorbate on the radiostability was studied. A comparison between the one-step labeling method and the two-step labeling method was another variable to consider, in regards to the preparation of the radioconstructs. In addition, the dependence of the in vitro binding of the radioconjugates, in vivo pharmacokinetics, tumor uptake on the mAb protein dose, and the heterogeneity of CAIX expression was studied in xenograft mice.

## 2. Results

### 2.1. Conjugation of p-SCN-βn-DOTA to cG250

DOTA(SCN)-cG250 immunoconjugates with a range of BFC/mAb loading ratios were prepared and purified by modulating the duration of the bioconjugation reaction times between 30 and 90 min. Immunoconjugates were classified by their molecular weight (MW) after conjugation with DOTA(SCN): C2 (90 min reaction time) > C3 (60 min reaction time) > C7 (30 min reaction time) detectable by size exclusion-high performance liquid chromatography (SE-HPLC/UV) and sodium dodecyl sulfate polyacrylamide gel electrophoresis (SDS-PAGE) (Figure 1). For example, conjugate C2 obtained after 90 min of incubation time showed a retention time of 12.24 min compared to 15.1 min for the native mAb; a shift of 3 min on the SE chromatograms. Likewise, conjugates C3 (12.72 min elution time) and C7 (13.66 min elution time) were incubated for 60 and 30 min, respectively, which correlated with the expected increment of the MW of the conjugates after the protein conjugation with the BFC was indicated by the left shifting of the peak on the SE-HPLC chromatogram, compared to the native cG250 (Figure 1a). The completeness of the protein purification by filtration–centrifugation in the conjugates was corroborated by monitoring the signal intensity within the 30 min region on the SE-HPLC chromatogram, which corresponded to the elution time of the free BFC. The complete removal of BFC in the purified conjugate is a critical parameter to ensure high radiochemical purity.

As expected for a denaturing analytical technique, SDS-PAGE chromatogram of the DOTA(SCN)-cG250 conjugates C2, C3, C4, C5, and C7 showed easily distinguishable bands corresponding to the light (LC) and heavy chain (HC) of the Abs. The conjugated protein bands of the LC and HC were, as expected, broader and shifted to higher molecular weights, when compared to the LC and HC bands of the native mAb (Figure 1c). Using a standard (Precision Plus™ Protein Unstained, 10–25 kDa, BioRad) the MW of the HC and LC bands of cG250 could be estimated. The results correspond to the same order of MW (C2 (90 min) > C3 (60 min) > C7 (30 min)) obtained by SEC chromatography.

For large-scale clinical production of immunoconjugates the reproducibility of the conjugation process is relevant. Consistency of the conjugation process was explored by undertaking the conjugation process in triplicate. Three conjugates (C3, C4, and C5) were prepared and purified by using a 60 min incubation time. The retention times of conjugates C3, C4, and C5 by SE-HPLC were similar, and the retention times of the conjugates varied by 12.64 ± 0.08% with less than 7% uncertainty (Figure 1b). In the case of the SDS-PAGE, the reproducibility obtained was less than 1% of uncertainty of the MW of the HC and LC (Figure 1c). Based on the SE-HPLC and SDS-PAGE results, the conjugates C3, C4, and C5 could be classified within the same group. The ratio of DOTA molecules per molecule of mAb, was calculated by the difference of the MW between the conjugated and the native mAb. The average ratio ranged from 1 to 6 in LC and from 4 to 23 in the HC based on the information from the SDS-PAGE chromatograms; using an uncertainty of 10% of the MW estimation by SDS-PAGE (Appendix A). Another important aspect that might influence the accuracy of the MW determination by SDS-PAGE is the preservation of the separation conditions. Different conditions such as the gel concentration and voltage during the protein separation would influence the migration of proteins in the tested sample, when compared to the standard sample, changing the standard mass calibration procedure as a consequence. In order to measure the MW with better accuracy, another analytical tool like mass spectrometry is required.

For SE-HPLC/UV, the stability of the conjugates was evaluated based on the shift between the retention times of the conjugates compared to the native mAb. This way, the variability of the retention time measurements at different days was excluded. These variations of retention time signals were observed due to the interactions of the stationary phase with the mobile phase (eluent or sample) components. After 24 months the difference between the retention times of the conjugates and the native monoclonal antibody were 2.91 ± 0.03, 2.51 ± 0.01, and 1.48 ± 0.03 for conjugates C2, C3, and C7, respectively. The intensity of the peak was the same using 0.05 mg/mL as protein concentration. The chromatogram of the conjugates showed no significant changes to peak positions or bands in SE-HPLC chromatograms or SDS-PAGE, respectively.

### 2.2. Identification of DOTA Modification Sites

The conjugates were analyzed through peptide-mass mapping using tandem liquid chromatography–mass spectrometry (LC–MS/MS). The modification sites by DOTA, were identified by comparing the peptide-mass mapping of the conjugates with the data base of peptide masses of the native cG250 after the digestion processes were completed with trypsin. The native cG250 sequence coverage LC–MS/MS was 96% and 88% for LC and HC, respectively. The data obtained was used as a control to identify the modified peptides in the conjugated protein and to calculate the occupancy by DOTA molecules on the amino acid sequence. The peptide mapping of the DOTA(SCN)-cG250 conjugates resulted in sequence coverages from 71% to 87% for HC and from 73% to 95% for LC, respectively. The lower coverage values were obtained in the conjugates with higher ratio DOTA(SCN) per molecule of mAb ratios. The higher the BFC/cG250 ratios led to a higher heterogeneity of the conjugated peptides, which resulted in decreased digestion efficiency of trypsin due to blocked trypsin cleavage sites, resulting in longer peptide sequences and consequently reduced extraction efficiency from the gel matrix. In order to increase the peptide sequence coverage, a second digestion process with chymotrypsin was performed for the conjugate C2. The combined data of two different digests resulted in almost 100% sequence coverage of both mAb subunits, HC (98.4%) and LC (98.6%). The combination of trypsin and chymotrypsin digests successfully increased the sequence coverage of the conjugates to 97.6% for the HC, and 92.5% for the LC. Short peptides could not be identified, which was due to chromatographic and data analysis/processing limitations.

Figure 2 shows that the DOTA(SCN)-modified peptides sites were, as expected, mostly through lysine amino acid residues (K and Lys), as previously described [25]. Also, DOTA modifications were not consistently allocated to the same K residues across all replicates on the HC and LC as shown by Figure 2a. The occupancy of K residues with DOTA(SCN) modification was calculated by MaxQuant, using peptide intensities and search parameter settings as previously for EasyProt. DOTA conjugated peptides were eluted at different retention times from the reversed phase chromatography column and detected with higher charge states than the nonconjugated peptides, as expected. DOTA(SCN)-labeled peptides eluted at different retention times from the reversed phase column than the native peptides, due to different matrix effects at the time of elution and possibly different ionization efficacies of native and conjugated peptide forms. The latter is supported by the observation that DOTA-labeled peptides were usually detected with a higher charge state than the nonlabeled peptides. We assumed that higher ratios represent a better accessibility of DOTA to the corresponding K residue.

K-416 of the HC appears to be almost quantitatively labeled, meaning that the probability of the conjugation of the K-416 is higher compared to the other K residues. Based on the incomplete sequence coverage achieved by in-gel digestion LC–MS/MS using trypsin enzyme, we also have to assume that there are K residues on peptides modified by DOTA(SCN), which could not be extracted from the gel matrix. Furthermore, DOTA(SCN) labeling renders peptides more hydrophobic, thus making it more difficult to extract them from the gel matrix. Therefore, nonlabeled peptides were extracted more efficiently than labeled ones, leading to an over-representation of the nonlabeled forms. It is well known and has been described that K is the most nucleophilic amine in proteins; however, K-416 has an additional input because it is the C-terminus of the HC of native cG250 [50,51]. The reaction that takes place does not have steric hindrance compared with the other K residues rendering the substitution reaction more efficient. In general, the reactivity of the N-terminal amino group is higher because its pKa value is lower than the K. Thus, DOTA(SCN) modifications were also observed at the amino terminus through aspartic acid amino residues (D, Asp) in the HC and LC.

### 2.3. Ratio of DOTA Molecules Per Molecule of Antibody

In general, when the average number of BFCs per protein (N) increases the immunoreactivity of the obtained product decreases. It is possible to determine “N” by calculating the difference in mass between the mass of the conjugates and the native mAb [52,53]. The mass of the conjugates was measured by matrix-assisted laser desorption/ionization time-of-flight mass spectrometry (MALDI-TOF MS) (Figure 2b) and the average number of DOTA(SCN) molecules per molecule of cG250 was then calculated. The native cG250 shows the presence of three major peaks corresponding to the MW of 148,736.14 Da (monocharged and unconjugated mAb, [M+H]^+^), 74,292.94 Da (doubly charged unconjugated mAb, [M+2H]^2+^), and 49,510.29 Da (triply charged unconjugated mAb, [M+3H]^3+^). Furthermore, the MALDI-TOF mass spectra of the DOTA(SCN)-cG250 conjugates also showed three peaks corresponding to the mono, doubly and triply charged conjugates species. Compared to the spectrum of the native cG250 mAb, these spectra show a broadening of the peaks, which indicates heterogeneity in the number and location of DOTA(SCN) molecules conjugated to the Abs, thus confirming the results of the LC–MS analysis. The MW of the peak [M+H]^+^ (151,543.63 Da) is lower for the conjugate C7 than for conjugate C3, corresponding to the results obtained by SE-HPLC/UV and SDS-PAGE (Figure 1). The MW of the conjugates C4 and C5 were also measured. The uncertainty of the MW by MALDI-TOF was also less than 1% in the three major peaks of conjugates C3, C4, and C5, which were prepared in the same conditions (data not shown). These results show a similar MW and, therefore, similar ranges of BFC/mAb ratios. The average number of DOTA(SCN) molecules per molecule of cG250 calculated for the conjugates are summarized in Table 1. Similar BFC per Ab ratios were obtained, by using the same conjugation method and incubation time but different mAbs [3,49].

In order to know the number of conjugated molecules of BFC in the HC and LC of the mAb, the native cG250 and conjugates were incubated with dithiothreitol (DTT) (40 mM) for 1 h before injecting the test sample in the MALDI-TOF spectrometer. The mass spectrum of the native cG250 showed peaks which corresponded to LC and HC. The mass spectrum of the reduced conjugate C3 also showed two main peaks at 51,546.27 Da and 24,151.98 Da, which corresponded to the MW of the HC and LC of the conjugated cG250, respectively. Those peaks were also broader compared to the spectrum of the reduced native cG250 (Appendix A). The average of DOTA(SCN) moieties was found to be 1–2 and 3–4 in the LC and the HC of conjugate C3, respectively (Table 1). The ratios were obtained per chain and were multiplied by two based on the antibody structure matched well with the ratios obtained for the nonreduced conjugate.

It was not possible to determine the ratio of BFC molecules in the HC and LC of conjugate C2 because it was not possible to measure the MW by MALDI-TOF, presumably due to the high heterogeneity of the conjugate. However, it was possible to determine the ratio of BFC molecules in the LC by intact mass using a QExactive mass spectrometer (ThermoFisher, Bremen, Germany) via a nanospray electrospray ionization (ESI) source. The samples (the native and the conjugated cG250) were treated with DTT to cleave the LC and HC. Very clean signals could be measured for the native LC and HC as well as the conjugated LC, while the conjugated HC spectrum was extremely complex and deconvolution resulted in several signals with an inherent uncertainty of being correct (data not shown). From the LC of conjugate C2, accurate mass determination was possible, however the HC presented challenges. The software was able to extract a few masses in the HC of the conjugate C2, but they were less certain than in all other measurements (95% vs. 99%). Using the obtained MWs of the native and the conjugate divided by the MW of the BFC (552.6 Da), the DOTA(SCN)/cG250 mAb ratio ranged from 5 to 6 in the LC. For the higher intensity MW peaks (50,682.105 Da and 50,519.625 Da) the average number of BFC ranged from 12 to 23 per mAb. Therefore, those results are not reliable because of the intensity of the rest of the peaks (background) on the chromatogram. The range of values corresponded to the ones obtained by SDS-PAGE with the range of the total MW of C2 obtained by MALDI-TOF (Table 1).

### 2.4. In Vitro Characterization of the Conjugates

Immunoreactivity of the conjugates was evaluated by flow cytometric and IHC analyses before labeling the cG250 with ^177^Lu. An increasing extent of conjugation of the cG250 with DOTA(SCN) moieties, correlates with a reduction in binding to CAIX on the SK-RC-52 cells compared to the native cG250 by flow cytometry and IHC (Figure 3). Staining of the cell lines with the native variant of cG250 showed that CAIX recognition on SK-RC-52 cell line is 20 times higher than on the SK-RC-18 cell line, which confirms the specificity of the cG250 and the suitability of these two cell lines for the remaining studies. The results were previously validated by immunocytochemistry analysis (ICC), where the SK-RC-52 cell line showed a strong staining intensity and cell membrane localization (Appendix A). However, CAIX expression was not detected in the SK-RC-18 cell line by flow cytometry and neither by immunocytochemistry (ICC) (Appendix A). 

Concentration-dependent experiments demonstrated an IC50 of around 0.02 µg/mL for cG250 (Figure 3a), which was used to evaluate the effect of DOTA(SCN) on the mAb recognition in subsequent experiments. At high doses (e.g., 100 µg/mL) of protein, the geometric mean fluorescence intensity (GMFI) values increase until the cG250 saturates all the binding sites. After conjugation of the cG250 with DOTA(SCN), a reduction in the recognition of the CAIX by the conjugates on SK-RC-52 cells compared to the native variant was observed (Figure 3b). The loss of recognition was dependent on the average number of conjugated DOTA(SCN) per molecule of mAb, as C2 contains more than 12–23 DOTA, C3 contains 8–10 DOTA, and C7 contains 3–5 DOTA. In order to explore if whether or not the location of the DOTA modifications had any influence over the loss recognition by the conjugated mAb, three different conjugates (C3, C4, and C5) with the same number of BFCs located in different K residues were compared. As evidenced in Figure 3c, there were no differences (*p* > 0.05) in the recognition between the three conjugates, indicating that their biological activity was directly related to the ratio BFC/mAb rather than the location of the BFC molecules on the K residues. Those conjugates were also analyzed by IHC and no significant difference between them was observed (Figure 3d).

The recognition of the conjugates to CAIX in tumor samples was another variable that was evaluated (Figure 3d). The native cG250 was evaluated showing a staining pattern that nicely reproduced the pattern of the commercial Ab, staining the same histological areas in frozen samples. CAIX staining with the C7 conjugate (conjugated with the lowest level of DOTA) was the most specific antibody without relevant background staining. The conjugate C2 showed a high background and no specific staining, which might be due to the high number of BFCs attached to cG250. In general, all conjugates showed the same staining pattern as the native cG250. The areas that did not express CAIX by the native and conjugated cG250 were comparable and showed the same pattern as the commercial Ab (Abcam), which was used as a positive control.

To evaluate possible recognition and accumulation of the radiolabeled cG250 in subsequent animal studies, the antigen was also measured in healthy tissue which was related to the metabolism and excretion of the large proteins. In the paraffin normal tissue samples, CAIX immunostaining, showed strong intensity in the stomach and negative immunostaining in the rest of the organs (liver, gallbladder, spleen, and duodenum) by Abcam Ab. However, normal tissue frozen samples were negative for cG250 mAb and the control Abs (Appendix A).

### 2.5. Radiochemical Purity and Influence of Sodium Ascorbate on the Stability In Vitro

Radioimmunoconjugates with different specific activities were obtained with >90% of radiochemical purity (RCP) by instant thin layer chromatography (ITLC) before and after the addition of DTPA. A slight excess of DTPA was added to the reaction mixture to complex any free ^177^Lu^3+^ ions, which was monitored on the SE radiochromatogram. After the purification processes by SE, the typical radiochemical yield of [^177^Lu]DOTA(SCN)-cG250 was 85% with radiochemical purity above 99%. It was possible to reach highest specific activities of 9 MBq/µg with the conjugate C3. In the case of conjugate C7 (less DOTA content), the RCP decreased when the initial activity was increased reaching as much as 5 MBq/µg. The presence of the ionic (free ^177^Lu^3+^) impurities is explained by the low DOTA content in this conjugate compared to the other conjugates. The average number of ^177^Lu atoms chelated per mAb was 1–2 for the conjugate C3 and 0–1 for the conjugates C7 postpurification. Higher initial activities of ^177^Lu were used to increase those values of specific activities of the labeled mAb, but they resulted in lower radiochemical yields.

Stability of the radioimmunoconjugates becomes a critical issue for conjugated mAb due to their pharmacokinetics and long circulation times. In vitro stability was studied under different conditions: HSA 20%, human serum (HS) and phosphate-buffered saline (PBS) with and without the addition of sodium ascorbate (NaAsc, 50 mg/mL) postpurification. Figure 4a shows a drop in the RCP of the radioconstruct two days after labeling at 319 MBq/mL activity concentration (specific activity was > 9 MBq/µg), even in the presence of the 50 mg/mL quenching solution. We assume that the protein was damaged to a certain degree, due to the formation of radical ions by radiolysis during the labeling process. Nevertheless, radiostability of [^177^Lu]DOTA(SCN)-cG250 at lower specific activities (<3 MBq/µg) and activity concentrations (<107 MBq/mL) was increased by the addition of NaAsc (Figure 4a). The RCP was above 90% by ITLC up to 10 d postlabeling in 20% HSA and HS for the three types of conjugates (Figure 4b).

### 2.6. In Vitro Radioimmunoreactivity of the Conjugates Labeled with ^177^Lu

During the radiolabeling process, the biological activity of the protein may be compromised and the specificity of the biomolecule to bind to CAIX antigen may therefore be reduced. Therefore, the immunoreactivity of the radioconstructs was evaluated in vitro using the SK-RC-52 cells before in the in vivo application. First, the specific binding to SK-RC-52 cells was measured using different protein concentrations and as a result, a maximum of 60% binding was obtained (Figure 5a). After adding 5 ng of carrier antibodies to 5 × 10^6^ cells, a decrease in the percentage of binding was observed due to the saturation of binding sites with the nonlabeled mAb. During the preparation of the radioconstructs it was not possible to separate the labeled-mAb from the nonlabeled-mAb using PD10 columns. Saturation of the binding sites was also observed by flow cytometric analysis (Figure 3a) at high protein doses. In the case of the SK-RC-18 cells (negative control) the percentage of binding and internalization was <2.0% (Figure 5a) as compared to the values for nonspecific binding using the native cG250 at the highest studied protein concentration (data not shown).

Figure 5b shows the percentage of binding and internalization of different conjugates to positive and negative CAIX cell lines (SK-RC-52 and SK-RC-18, respectively). The percentage of binding of [^177^Lu]DOTA(SCN)-cG250 to SK-RC-52 cells ranged from 60 to 70% for conjugate C7 with a ratio of less than six DOTA(SCN) molecules per cG250. The percentage of binding to the SK-RC-52 cell line was significantly higher (*p* < 0.001) for the radioconstructs using conjugate C7 (lowest DOTA content). The percentage of internalization of the radioconstruct was >90% of the total activity bound to the SK-RC-52 cells (Figure 5b,c). In general, the percentage of binding and internalization decreased with increasing DOTA content showing the same pattern as observed in the flow cytometric analysis (Figure 3b). Figure 5c shows no significant change (*p* > 0.05) in the percentage of binding and internalization after a dilution of the radioconstructs with 20% HSA, human serum, and PBS to simulate in vivo conditions. The percentage of binding and internalization slightly decreased using higher specific activities of the radioconstructs (Figure 5d). Blocking studies with an excess of native cG250 revealed the specificity of the labeled cG250 by reducing binding and internalization to background levels. In all cases, the SK-RC-18 cells were used as a negative control to correct the percentage of binding and internalization as a nonspecific binding.

### 2.7. Biodistribution

The biodistribution of [^177^Lu]DOTA(SCN)-cG250 which was measured 48 h after the I.V. application of 12 MBq in BALB/c nu/nu mice across a range of Ab protein mass doses (specific activities) shows a moderate influence by the total applied protein dose (Figure 6). In vivo, blood retention increased with higher protein doses (i.e., lower specific activity) and liver uptake slightly declined with protein doses up to 60 µg/animal. In contrast, tumor accumulation was not significantly influenced (*p* > 0.05) by total protein doses between 5 and 100 µg per animal. Correspondingly, at higher protein doses blood circulation seemed prolonged and tumor uptake was highest at a relatively high amount of cG250 (30 µg) per animal (Figure 6a). In addition, a higher amount of radioactivity remained in the rest of the animal body (skin, blood, muscles, and bones) for animals with higher administrated protein doses and the excretion process was low (Figure 6b). Longer circulation and the remaining activity in the rest of the animal may be correlated to the saturation of the binding sites in the target, which was observed during the blocking studies. A reduction of tumor uptake at 24 h (*p* < 0.05) was observed in the animals with a previous injection of 500 µg of native cG250 (Appendix A). The native mAb was injected shortly before the radioconjugate and CAIX receptors where partially blocked. However, a major impact by the blocking in spleen and liver was not observed.

Tumor volume had more influence on tumor uptake when low protein doses were used (Figure 7). Interestingly, for one animal bearing a small tumor (140 µg), a positive out layer was observed in the group receiving the lowest protein dose. In case of higher doses, the %ID/g was similar across mass doses and was independent of the tumor size (Figure 7). The effect of the tumor weight on tumor retention has been studied in previous work, where an exponential decline of tumor uptake of ^111^In-DTPA-girentuximab was demonstrated [54].

The relationship of blood circulation values with the protein amount was also observed in the biodistribution of [^177^Lu]DOTA(SCN)-cG250 measured 24 h and 96 h after the I.V. application of 2 MBq (0.5 µg) and 18 MBq (5 µg) without adjusted doses (Figure 8). The animals with a 5 µg protein dose showed a slower blood clearance until 24 h. The T/B ratio of 43.25 at 96 h was almost four times higher than the ratio at 24 h in the animals with a 5 µg protein dose. Therefore, tumor uptake was higher in the animal group with lower liver uptake. The decreased %ID/g in the liver with an increased protein dose was observed independently regardless of if the protein dose was adjusted (Figure 6 and Figure 8) [33,55].

In order to determine the availability of the radioconstruct in blood and in vivo stability, a metabolism analysis was performed by protein precipitation (data not shown). The percentage of the activity was >50% in plasma showing a low complexation with blood cells. Another important parameter is the percentage of free ^177^Lu activity in blood plasma relative to the intact labeled radioconstruct. The percentage of free ^177^Lu was <5% at 24 h and increased to 10% at 96 h from the release of the radiometal. Indeed, the tumor/blood (T/B) ratio increased at 96 h (0.5 µg) to 10.16 (Figure 8a) when the specific activity was <3 MBq/µg. This could indicate the importance of the radiostability of the radioconstructs in vivo. Therefore, tumor uptake was low but in line with the values obtained by Muselaers, C. H. (escalating doses of ^111^In-DTPA-G250) comparing the lower protein dose (<5 µg) [56].

### 2.8. Effect of Tumor Volume, Hypoxia, and Necrosis on cG250 Tumor Retention

The accumulation of the radioactivity in the tumor is also related to the properties of the target such as the location, size, antigen expression, and others. To study the target influence in the radiopharmaceutical uptake and diffusion, tumor properties such as CAIX expression and the vascularity were studied in different-sized tumors. IHC analysis of SK-RC-52 tumors (*n* = 20) showed a CAIX expression depending on the tumor size and tumor growth. In general, it resulted in decreased availability of the antigen when the tumor volume was increased. Tumors smaller than 20 mm^3^ (Figure 9a) and tumors larger than 200 mm^3^ (Figure 9c,d) usually developed necrosis after 6 weeks of tumor growth. The extent of necrosis consequently reduces the expression of CAIX. Large tumors (544 mm^3^) showed 80% necrosis (as observed by H&E stain) (Figure 9c). However, tumor volumes smaller (161 mm^3^) showed mostly homogeneous patterns and a strong intensity of CAIX localized in cell membranes of the SK-RC-52 tumors and no necrosis (Figure 9b).

The vascular density was determined by CD31 (a marker of endothelial cells) (Figure 9e–h). Additionally, CD31 staining showed an average percentage of vascular density of 2.5 ± 0.5% in the tumors, which was also independent of the tumor size and the CAIX expression. Results are in line with the values obtained by Oosterwijk-Wakka, J.C. in 2015 [57]. Hypoxia-inducible factor 1 (HIF-1) marker, which is a key mediator of tumor survival and adaptation to a hypoxic environment, was evaluated by IHC [58]. IHC analysis of the SK-RC-52 tumors with different size showed the presence of HIF1α in all nuclei from tumor cells (Figure 9i–l) but also reflects necrosis and a high heterogeneity of CAIX expression across the tumor. These effects have a direct influence on the targeting of the mAb and resulting radiation dose accumulation to the tumor [59].

## 3. Discussion

In the present study we evaluated some variables that might affect the immunoreactivity of the cG250 during the conjugation and radiolabeling processes in comparison with the initial properties of the native cG250. DOTA isothiocyanate modifications occurred via K residues with the formation of a stable and strong thiourea bond (SC(NH_2_)_2_). The precision of the isothiocyanate conjugation to mAbs via lysine residues allows for higher selectivity and control compared to the iodination reaction. Higher levels of modification can still lead to impaired binding, and, therefore, loss of efficacy. However, the extent of Ab modifications via BFC/mAb ratio can be controlled during the bioconjugation reaction. The absence of DOTA(SCN) labeling in the CDR is an additional advantage of the labeling of mAb with radiometals compared to previous studies with ^131^I [55]. The radioiodination of cG250 using the standard method Chloramine–T is performed through tyrosine amino residues (Y, Tyr), which are part of the complementarity determining regions (CDRs) of the cG250. In in vivo studies the bond ^131^I-Y amino residues might be affected due to the action mechanism of the mAb during the internalization process into the cancer cells [60]. The use of the radiometals is a tool more suitable for RIT of RCC [2]. However, the occupancy of the K in the neighborhood of the CDRs can compromise the biological activity of conjugates and limit the effectiveness of the therapy. The higher the number of DOTA(SCN) attached to K residues in the mAb sequence, the more heterogeneity the conjugate has and the more difficult it is to characterize. For example, conjugate C2 (90 min) was impossible to measure the MW and calculate then the BFC/mAb ratio by mass spectrometry.

The effectiveness of RIT depends on a number of factors and processes. Some factors relate to the specificity, affinity, and immunoreactivity of the mAb postconjugation and post-radiolabeling. The in vitro evaluation of the immunoreactivity of the conjugates showed that the conjugates with the lowest DOTA content have a better recognition of the CAIX compared to conjugates with higher DOTA content using the native cG250 as a reference. Meanwhile, conjugates with the same BFC/mAb ratio showed similar percentage of binding to the CAIX antigen in SK-RC-52 cells and tumor samples, which was observed by flow cytometry and immunohistochemistry. The immunoreactivity of the DOTA(SCN)-cG250 conjugates seemed mainly determined by the average number of the BFC attached to the mAb. Also, the introduction of multiple BFC modifications might enhance blood clearance and thus the deterioration of pharmacokinetic properties occurred [29]. Another critical point for the use of radiolabeled mAbs is stability and immunoreactivity after the labeling or chelation with the radiometal. The radiochemical yield of DOTA(SCN) conjugates at room temperature in general, is very low; but offers advantages over heating most importantly, such as the preservation of the protein. To complete the radiolabeling (RCP > 90%) without loss of immunoreactivity of the radiolabeled Abs, longer reaction times (90 min) and elevated temperatures (below 42 °C) were used. We were able to label DOTA(SCN)-cG250 conjugates with ^177^Lu obtaining RCP > 95% for each conjugate (different BFC/mAb ratios). Compared to a two-step method the ratio of ^177^Lu to mAb was orders of magnitude higher. As the effectiveness of the RIT also depends on the accumulation of radioactivity at the tumor site, the specific activity of the radioconstruct might play a determining role in accessible tumor sites such as circulating tumor cells and tumor cell clusters. With prevention of radiolysis it was possible to keep the stability in vitro in physiological conditions for specific activities < 3 MBq/µg (<107 MBq/mL) by adding sodium ascorbate to the formulation after labeling. In general, the immunoreactivity of the radiolabeled cG250 to the specific binding of SK-RC-52 cells decreased when the DOTA content per conjugates was high. The internalization and accumulation of the radionuclide into the target is an important parameter in RIT [24], and does not seem significantly influenced by the increment of DOTA modifications. However, the percentage of internalization to the SK-RC-52 cells was not significantly different (*p* > 0.05) between conjugate C3 (8–10 BFCs/mAb ratio) and conjugate C7 (5–6 BFCs/mAb ratio).

The biodistribution of [^177^Lu]DOTA(SCN)-cG250 in Balb/c nu/nu mice with subcutaneous SK-RC-52 showed more than 20% ID/g in the liver and less than 10% ID/g in the tumor, independent of the protein dose which ranged from 5 to 100 µg. Both lower as well as higher tumor uptakes were observed when increasing the protein dose. The blood clearance appeared generally slightly faster with lower amounts of protein. Typically, the therapeutic index between the tumor effect and systemic toxicity is determined by accumulation and retention within tumor tissue on one side and blood circulation on the other. Due to the possibility of slow accumulation and further internalization into the tumor tissue, long blood circulation might enhance both tumor accumulation and systemic toxicity. Thus, a complex in vivo interaction between pharmacokinetic (PK) properties of the radioconstruct and tissue properties of the tumor exist. For smaller molecules like ^99m^Tc-(HE)3-ZCAIX:1 the influence on PK on tumor uptake is generally less pronounced [61]. At very low protein amounts (0.5 µg girentuximab/animal) rapid clearing from blood was observed, which results in a shorter time of availability of the radioimmunoconjugate for internalization into the tumors and correspondingly very low tumor uptake. At high mAb doses oversaturation of binding sites occurs and tumor uptake is obviously maximized mainly by the possibility of long diffusion into the tumor and by resaturation of recycled antigens after internalization. Although higher specific activities are very feasible, the lowest amount of cG250 showing acceptable PK was 5 to 30 µg/animal. This resulted in typical tumor accumulation of 10% ID/g in tumors with higher uptake in small and homogenous tumors. This compares well to tumor uptake of F(ab’)2 fragments in head and neck squamous cell carcinomas (HNSCC) xenografts of 1 to 4% ID/g [45]. Nevertheless, tumor accumulation is highly influenced by the heterogeneity of tumor microstructure. However, at much lower protein doses, the PK might be an obstacle when the radiolabeled Ab is rapidly cleared from the blood and distributed to organs like the liver and spleen. Thus, PK of [^177^Lu]DOTA(SCN)-cG250 (3 MBq/µg) was influenced by the applied total protein dose, whereby at low amounts of mAb rapid clearance from blood by to liver and spleen occurred.

For disseminated small tumors, including single cancer cells, diffusion is less important than saturation of antigens. This is particularly true for systemic disease-spreads where tumor cells and tumor cell clusters are distributed through the body. Here, a high ratio between radioactivity and the amount of protein carrier mAbs might be beneficial. When binding occurs in relation to the circulation time, it is plausible to measure a relationship between tumor accumulation and tumor size. However, if oversaturation occurs, such a relationship declines when binding sites stay saturated over the blood clearance time. Subsequently, the expected inverse relationship between tumor size and tumor uptake was not observed at protein doses of 60 and 100 µg per mouse. This indicates that protein amounts above 60 µg will not be optimal for binding, in particular, for small tumors and single cells. To target subclinical tumor manifestations—either single cells or small tumor cell clusters—replacement of a beta emitting isotope by long lived alpha emitting isotopes might be promising [4,5,62,63]. Furthermore, the effect of longer blood circulation with a higher protein amount is evident with a higher inverse relationship between the tumor size and tumor targeting for animals receiving 5 µg, over animals receiving only 0.5 µg cG250 without adjusting for specific activity. Therefore, future development of targeting smaller tumor clusters without the observed tendency to necrosis and heterogeneity might include alpha emitting isotopes like Actinium-225.

Other factors related to the target or antigen such as density, location, and heterogeneity of expression of tumor-associated antigen within tumors will affect the therapeutic efficacy of RIT, as will physiological factors such as the tumor vascularity, blood flow, and permeability [10,59]. Consistent with decreased vessel density, elevated intratumoral pressure, and the presence of necrotic areas, large tumors only display a very low uptake. A minimum value of tumor–antigen density is a prerequisite for mAb/ADC efficacy [11]. In view of tumor properties influencing radiopharmaceutical uptake and diffusion, great heterogeneity with regards to the occurrence of necrosis and expression of CAIX was detected. Those properties were studied in different sized-tumors and resulted in decreased availability of the antigen when the tumor volume was large. In addition, the poor vascularization could influence the reachability of the target and thus the tumor uptake.

We show that, the transportation of therapeutic activities to the tumor tissue by cG250 was influenced by several factors. Thus, an optimization of the radiolabeling and BFC/mAb ratio along with an optimized protein dose is important to be able to preferentially target small and homogeneous CAIX expressing tumors. An inverse relationship between the tumor volume and tumor uptake was found. Tumors smaller than 20 mm^3^ and larger than 200 mm^3^ mostly developed necrosis after six weeks of tumor growth. In addition, after four weeks of tumor growth variation in the tumor volume increased because the growth of the tumor size among the animals was very different. This resulted in dispersed %ID/g values in the animal experiments and resulted in the interpretation of the results being difficult. Heterogeneous expression of the CAIX antigen and necrosis resulted in lower tumor uptake. In particular, large tumor sizes are not ideal targets. However, investigations targeting small tumor cell clusters are typically not directly possible, due to the difficulties of detection within the organism. Therefore, we conclude that specific activities of 2–10 MBq/ug and antibody amounts of 5–30 µg per mouse are optimal to investigate [^177^Lu]cG250-mediated therapies regarding differently accessible target cells and to optimize repeated application schemes.

## 4. Materials and Methods

### 4.1. Preparation of cG250 Conjugates

cG250 was provided by the company Wilex (Munich, Germany). Isothiocyanate-benzyl-DOTA (p-SCN-βn-DOTA) was purchased from Macrocyclics (Dallas, TX). cG250 (5 mg) was mixed with *p*-SCN-Bn-DOTA (2.5 mg) in 50 µL of sodium bicarbonate buffer (1 M NaHCO_3_) following the previously described protocol [49]. The mixture was incubated at 37 °C for 30, 60, and 90 min using a thermomixer to obtain conjugates with different BFC molar ratios per molecule of mAb. The mixture was purified by filtration–centrifugation using a Millipore centrifugal device YM-10 (10,000 MW cut off, Millipore). The sample was centrifuged (4000 *g* × 5 min) at 4 °C to remove the unreacted BFC and buffer exchange the conjugation into a saline solution (NaCl 0.9%) using five volumes of NaCl 0.9%. Postpurification, the samples were transferred to Eppendorf vials (2 mL) previously weighted and labeled C2 (90 min), C3–C5 (60 min), and C7 (30 min). The concentration of the conjugates were determined by ultraviolet spectroscopy (UV) using 1.4 mL/mg/cm as extinction coefficient (Nanovue-GE Healthcare Life Sciences).

The purified fraction was analyzed by SE-HPLC/UV using a G3000 column (7.5 × 300 mm, 10 µm, Waters). The samples were prepared in triplicate for each of the conjugates (0.05 µg/mL) and measured at 280 nm with a flow rate of 0.5 mL/L of 0.9% NaCl (Dionex, P680 HPLC pump, UVD1704 detector). The spectra were evaluated by the software Chromeleon from Dionex (Chromeleon 6.8 SR13 Build 3967 Version). The native cG250 was used as a reference in all cases. The elution time of the peaks of the conjugate were compared with the native cG250.

The conjugates were also analyzed by electrophoresis SDS-PAGE using 10% gel. The native cG250 and conjugated samples were incubated with a sample buffer (0.125 M Tris-HCl pH 6.8, 4% (*w*/*v*) SDS, 10% (*v*/*v*) glycerol, 0.01% bromophenol blue, 40 mM DTT) and heated at 95 °C for 5 min following the protocol by Laemmli [64]. The electrophoresis was run at a constant voltage of 100 mV (Hoefer SE250 Mini-Vertical gel Electrophoresis unit, USA). The visualization of the light chain (LC) and heavy chain (HC) bands were performed by Coomassie blue G-250 al 0.05% (Merck) blue-stained [65] and analyzed with ChemiDoc XL Imager (Bio Rad) by Image Lab Software 5.2.1 (Bio Rad). Precision Plus™ Protein Unstained (10–25 kDa, BioRad) was used as a MW marker.

### 4.2. Characterization by Mass Spectrometry

Prior to the mass spectrometric identification by LC–MS/MS, HC, and LC bands were excised and in-gel digested to generate the peptides as previously described [66]. Extracted peptides were loaded onto a precolumn (PepMap C18, 5 µm, 300 A, 300 µm × 15 mm length) at a flow rate of 20 µL/min with solvent A (0.1% formic acid in water/acetonitrile 98:2) with an Ultimate-3000 (ThermoFisher, Reinach, Switzerland), thereafter eluted in back flush mode onto the analytical nanocolumn (C18, 5 µm, 300A, 0.075 mm i.d., × 150 mm length) using an acetonitrile gradient of 5 to 40% of solvent B (0.1% formic acid in water/acetonitrile 4.9:95) in 40 min at a flow rate of 400 nL/min. The column effluent was directly coupled to a Fusion LUMOS mass spectrometer (ThermoFischer, Bremen; Germany) via a nanospray ESI source. Data was acquired in data-dependent mode with precursor ion scans recorded in the Orbitrap at a resolution of 120,000 (at *m*/*z* = 250), which is parallel to the top speed of the fragment spectra of the most intense precursor ions in the linear trap, for a maximum cycle time of 3 s. Peptides were fragmented in parallel by high-energy collision and electron transfer.

The fragment spectra acquired by LC–MS/MS were converted to a mascot generic file format (mgf) by ProteomeDiscoverer 2.0 (ThermoFisher Scientific) and interpreted with Easyprot (version 2.3) searching against the SwissProt human protein sequence database (version 2014_01) including the sequence of native cG250 using fixed modification of carbamidomethylation on cysteine (Cys, C), variable modifications of oxidation on methionine (Met, M), deamidation on glutamine (Gln, Q)/asparagine(Asn, N), and DOTA on K residues and on protein N-Terminus, respectively. A forward + reversed sequence database search was used to estimate the false discovery rate. Parent and fragment mass tolerances were set to 10 ppm and 0.4 Da, respectively. Protein identifications were only accepted, when two unique peptides fulfilling the 1% false discovery rate (FDR) criteria were identified. Furthermore, the occupancy of K residues with DOTA modification were calculated by MaxQuant (version 1.5.0.0) using peptide intensities and the search parameter settings as used for EasyProt. Mass spectrometry sequencing data was acquired at the Proteomics and Mass Spectrometry Core Facility, Department for Biomedical Research (DBMR), University of Bern, Switzerland.

The determination of the MW and the average number of BFC groups attached to each mAb molecule was performed by MALDI-TOF-MS. Samples were injected on an Autoflex III Smartbeam instrument (Bruker Daltonics, Bremen, Germany). An analysis was performed in the linear mode with a positive polarity, and a mass range of 200 kDa. Conjugated cG250 solution was diluted (1:10) with saturated α-cyano-4-hydroxycinnamic acid (CHCA, MW 189.04 Da) solution. Data acquisition and processing was performed by flexAnalysis software (Built 75, version 3.3). MALDI-TOF mass spectra data was obtained at the mass spectrometry group, Department of Chemistry and Biochemistry, University of Bern, Switzerland.

### 4.3. Cell Culture

The human kidney carcinoma cell lines, SK-RC-52 and SK-RC-18, were kindly provided by Dr. Weis-Garcia from Memorial Sloan Kettering Cancer Centre (MSKCC, New York, USA). SK-RC-52 is a CAIX-expressing human RCC cell line derived from mediastinum and SK-RC-18 is derived from lymph node cells, negative for CAIX [39]. The cells were cultured in RPMI medium (Life Technologies) supplemented with 10% fetal calf serum (FCS), 100 IU/ml of penicillin and 100 µg/ mL of streptomycin at 37 °C in a humidified atmosphere with 5% CO_2_.

### 4.4. Flow Cytometric Analysis

The Abs were briefly diluted in 100 µL of fluorescence-activated cell sorting (FACS) buffer (Dulbecco’s phosphate buffered saline with 0.2% bovine serum albumin, BSA) in concentrations ranging from 3 ng/mL to 100 µg/mL, or as otherwise indicated. The mAb dilutions were mixed with 10^5^ cells and incubated for 60 min at 37 °C. Subsequently, the cells were washed three times with a FACS buffer and incubated with a R-Phycoerythrin AffiniPure F(ab’)_2_ Fragment Goat Anti-Human IgG + IgM (H + L) (Jackson Immunoresearch) for 30 min at 4 °C in the dark. Next, the samples were washed and resuspended in the FACS buffer for analysis. An IgG1 from myeloma (Sigma) was used as an isotype control to define the threshold of the background staining. The analysis was carried out on a FACSVerse (BD Biosciences, San Jose, CA, USA). The data was analyzed by FlowJo (Tree Star, Ashland, OR, USA). The flow cytometry analysis was performed at the Institute of Pharmacology (PKI), University of Bern, Switzerland.

### 4.5. Radiolabeling, Quality Control, and Radiostability

DOTA(SCN)-cG250 conjugates were labeled with ^177^Lu (>500 GBq/mg, IDB Holland) in an ammonium acetate buffer, pH 5.5–6.5, at 37 °C for 90 min as described previously [49]. All solutions used for the labeling were prepared with metal-free water and filtered using 0.22 µm filters (Millipore). Conjugates with different specific activities were obtained by labeling 100 µg of protein with ^177^Lu activities (0.05 M HCl) from 0.1 to 3 GBq. The reaction was stopped after 90 minutes by adding diethylene triamine pentaacetic acid (DTPA, 10 mM) and incubation at 37 °C for another 30 min. The radioconstructs were purified by size exclusion (SE) using a PD 10 column and 1% HSA in PBS. ITLC was performed using a silica gel (SG 1.5 × 10 cm, Varian) as stationary phase and 20 mM of DTPA as a mobile phase (Raytest, MiniGita Firm, ver. 1.10). The chromatograms were evaluated by Gina Star TLC software (ver. 5.0.1 Rel 2). Purified radioconstructs were diluted with human serum, 20% of HSA and PBS in order to simulate in vivo conditions. The stability of the radioconstructs at different specific activities was studied in the presence or absence of sodium ascorbate at 37 °C.

### 4.6. Radioactivity Binding Assay

The binding assay was performed in triplicate using 200 µL of SK-RC-52 and SK-RC-18 cells. As a control, nonspecific binding was carried out incubating 500 µg of the native cG250 with the SK-RC-52 cell for 5 min before the addition of the radioconstruct. The radiocontructs were diluted in HSA/PBS to 5 × 10^−2^ µg/mL and 10 µL of the radiolabeled mAb was added to the cells and incubated at 37 °C for 1 h. After incubation, the cell pellets were washed twice with 1% HSA/PBS, centrifuged (400× *g*, 5 min) and measured by an automatic gamma counter (WIZARD2, PerkinElmer). The pellets were then treated with a stripping buffer (0.1 M CH_3_COOH, 0.15 M NaCl, PBS, pH = 3) to determine the percentage of internalization.

### 4.7. Animals

3 × 10^6^ SK-RC-52 cells in 100 µL PBS were injected subcutaneously into the right flank of 6–8 week old male BALB/c nu/nu mice (Janvier, le Genest-Saint-Isle, FR). Two different dimensions (length and width) of tumor size were measured at least twice a week with a caliper. The tumor volume was calculated using the equation: V = (π/6)*(higher diameter)*(lower diameter)^2^. All animal experiments were performed in accordance with German law and guidelines for care and use of laboratory animals. The experiments were approved by the competent authority (Landesuntersuchungsamt Rheinland-Pfalz, Germany; according to §8 Abs. 1 Tierschutzgesetz; permission no. 23177-07/G15-1-033).

### 4.8. Immunohistochemical Analysis (IHC)

Tissue samples of the stomach, liver, gallbladder, spleen, and tumor were collected and divided into two parts. One part of the tissue sample was fixed in a formaldehyde solution at 4% and embedded in paraffin (FFPE). The other remaining tissue sample was embedded in OCT (Tissue-Tek^®^) and frozen at −80 °C. The samples were transported to the Translational Research Unit (TRU) at the Pathology Institute, University of Bern, Switzerland to perform the immunohistochemical analysis (IHC). The samples were cut to 3 μm thickness and IHC analysis was carried out in the automated system BOND RX (Leica Biosystems, Newcastle, UK). FFPE sections were deparaffinized and rehydrated in a dewax solution (Leica Biosystems) and the antigen was retrieved by heating in a citrate buffer solution (pH = 6.5) at 95 °C for 20 min. Endogenous peroxidase activity was blocked with a H_2_O_2_ solution for 4 min. Carbonic anhydrase rabbit polyclonal Ab (Abcam, ab15086) was incubated at 1:1500 dilution for 30 min at room temperature. This Ab was used to validate the native and conjugated cG250. Hypoxia-inducible factor 1 (HIF-1) rabbit polyclonal Ab (Genetex, GTX127309) was diluted at 1:1000 and incubated for 30 min to confirm the hypoxia conditions into the tumor. The CD31 rabbit polyclonal Ab, (Abcam, ab28364) was diluted at a 1:30 dilution and incubated for 2 h to detect the endothelial cells from blood vessels to show extent of vascularization of the tumors. The frozen samples were fixed in acetone for 15 min at −20 °C and then air-dried. The samples were incubated with native and conjugated cG250 at a 1:1000 dilution for 30 min and subsequently the secondary rabbit anti-human Ab (Dako, P0214) was used at a 1:400 dilution for 15 min. As a negative control, the same tissue samples were incubated in parallel with only a secondary rabbit anti-human Ab.

All the samples were visualized with the Bond Polymer Refine Kit with 3-3′-Diaminobenzidine-DAB as the chromogen (Leica Biosystems). The samples were then counterstained with hematoxylin and mounted in Aquatex (Merck, Darmstadt, Germany). Finally, all slides were scanned and photographed in a Pannoramic P250 scanner (3DHistech, Hungary). By using ImageJ, the vascular density through CD31 expression was quantified for each tumor sample as a percentage of the CD31-positive microvessel area to the total tumor area (CD31 area/total tumor area) as previously described [67].

### 4.9. Biodistribution

After 4 to 6 weeks of tumor growth, groups of four mice were injected intravenously (I.V.) via the tail vein with 12 MBq of [^177^Lu]DOTA(SCN)-cG250 with adjusted protein doses with native cG250 from 5 to 100 µg. The animals were sacrificed at 48 h post injection and the organs were collected. To determine differences in pharmacokinetics and its influence on tumor uptake at very low protein doses (n, groups of four animals were injected with 0.5 µg (2 MBq) and 5 µg (18 MBq) at a fixed specific activity of 3 MBq/µg and biodistribution was determined after 24 h and 96 h. Blood samples were collected, mixed with a 100 µL heparin solution to avoid coagulation, weighed, and the activity measured with a gamma counter. After the addition of 500 µL of PBS, the blood was centrifuged at 7500 rpm for 5 min to separate blood cells and plasma. The plasma fractions were weighed and measured with a gamma counter. Proteins were precipitated by adding 500 µL of acetonitrile and separated by centrifugation at 7500 rpm for 5 min. The supernatants were weighed followed by the determination of the activity with a gamma counter. The percentage of radioactivity in the blood cell was calculated by subtracting the activities of the supernatant from the activity of the whole blood. In all the experiments the percentage of incorporated doses (%ID/g) was calculated by the weight of the organs and the measurements by gamma counter (WIZARD2, PerkinElmer).

### 4.10. Statistical Analysis

All statistical analyses were performed using the program GraphPad Prism (version 6.0). The one-way ANOVA was used for the comparison between the conjugates by FACS, and radioactivity curves and blocking in vivo experiments. Two-way ANOVA was used to compare the groups at different specific activities and different stability conditions. Differences were considered to be statistically significant at a level of *p* < 0.05. The %ID/g in the biodistribution studies were presented with the average of the same group of animals and their standard deviation.

## Figures and Tables

**Figure 1 pharmaceuticals-11-00132-f001:**
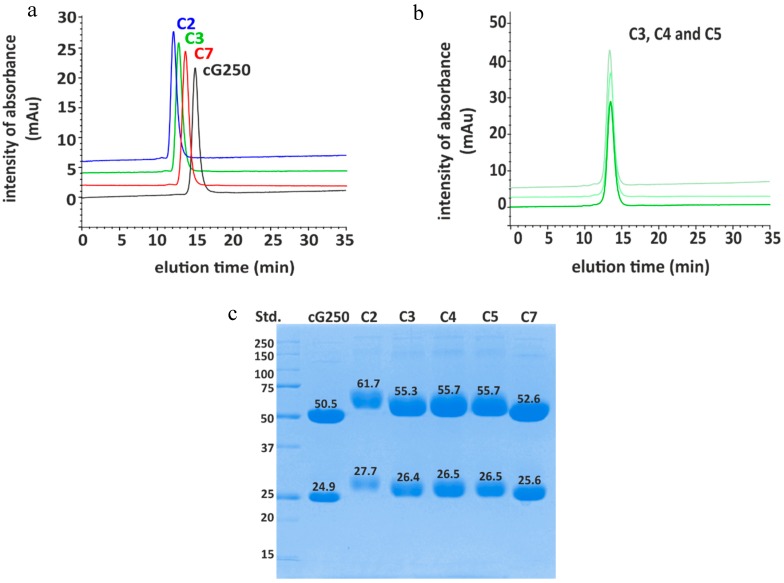
Conjugation of cG250 with DOTA via benzylthiocyano group: (**a**) Size exclusion-high performance liquid chromatography (SE-HPLC) chromatograms of different selected DOTA(SCN)-cG250 conjugates compared to the native cG250; (**b**) size exclusion-high performance liquid chromatography/ultraviolet (SE-HPLC/UV) chromatograms of DOTA(SCN)-cG250 conjugates prepared from the same batch; and (**c**) SDS-PAGE chromatograms of cG250-conjugates in reduced conditions.

**Figure 2 pharmaceuticals-11-00132-f002:**
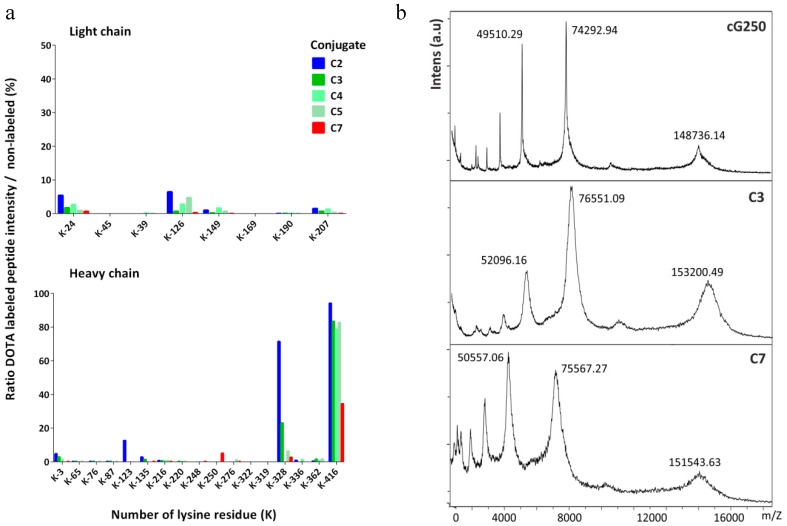
Characterization of the DOTA(SCN)-cG250 conjugates by mass spectrometry: (**a**) Lysine occupancy in the LC (top) and HC of DOTA(SCN)-cG250 post trypsin in-gel digestion of the proteins (*n* = 3) in conjugate C2 (90 min), C3, C4, C5 (60 min), and C7 (30 min) and (**b**) mass measurements by MALDI-TOF MS of native cG250 and conjugates C3 and C7.

**Figure 3 pharmaceuticals-11-00132-f003:**
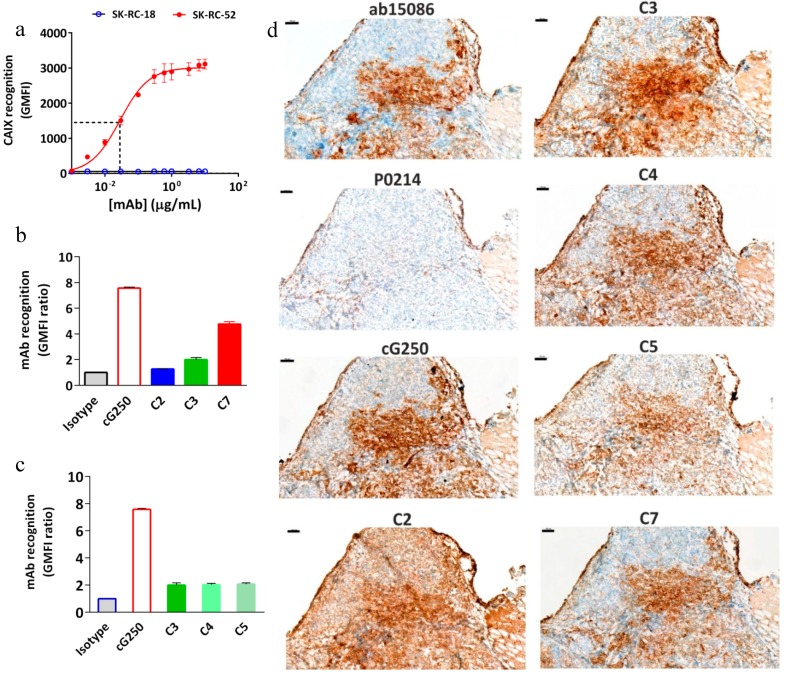
Immunoreactivity of DOTA(SCN)-cG250 conjugates with SK-RC-52 cells in vitro and in tumor tissue. (**a**) Concentration-dependent binding of native cG250 to SK-RC-52 cells and SK-RC-18 (control), as assessed by flow cytometry. The red dashed line corresponds to the IC50. (**b**) Flow cytometric assessment of CAIX recognition on SK-RC-52 cells of immunoconjugates C2 (90 min), C3 (60 min) and C7 (30 min). (**c**) Flow cytometric assessment of CAIX recognition on SK-RC-52 cells of immunoconjugates C3, C4, and C5 (60 min). (**d**) CAIX immunostaining in frozen SK-RC-52 tumor samples using native cG250 mAb and immunoconjugates C2, C3, C4, C5, and C7. CAIX Abcam (ab15086) as a positive control and Dako (P0214) as a negative control. 100 μm scale bar and 10X objective.

**Figure 4 pharmaceuticals-11-00132-f004:**
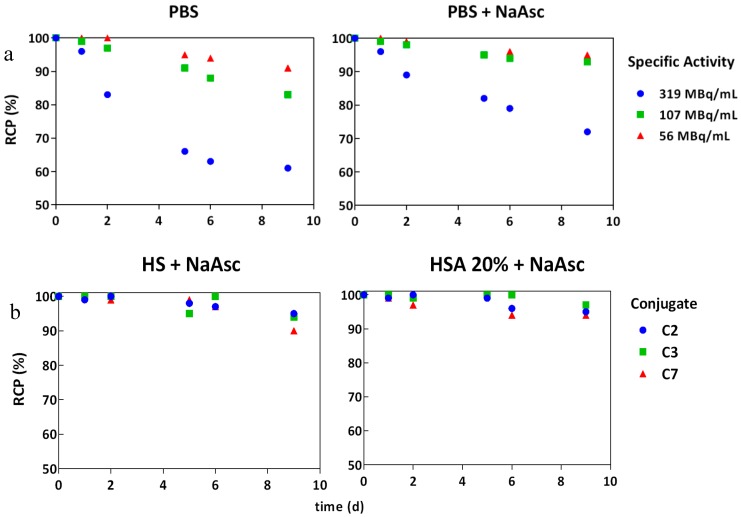
In vitro stability of [^177^Lu]DOTA(SCN)-cG250 in the presence of NaAsc by TLC: (**a**) C3 radioconstructs at different activity concentrations without and with NaAsc respectively and (**b**) C2, C3, and C7 radioconstructs (2 MBq/µg) in human serum and HSA 20% in the presence of NaAsc.

**Figure 5 pharmaceuticals-11-00132-f005:**
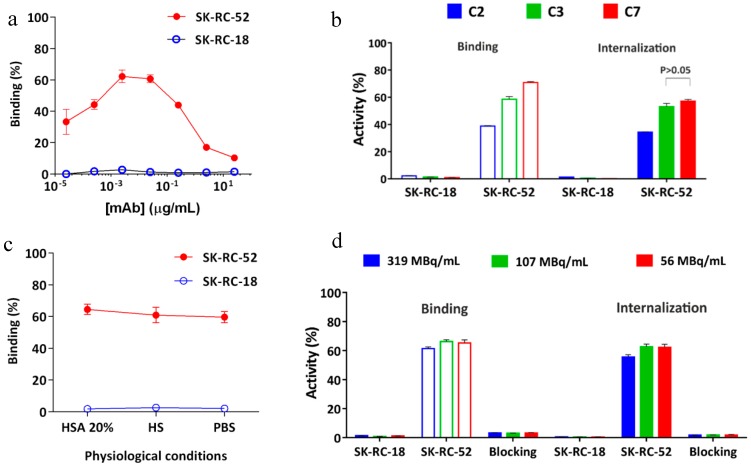
Radioimmunoactivity in vitro of [^177^Lu]DOTA(SCN)-cG250 to SK-RC-52 cells (*n* = 3). (**a**) Concentration-dependent binding of [^177^Lu]DOTA(SCN)-cG250 using the conjugate C3 (60 min). (**b**) Recognition of radioconstructs from conjugates C2 (90 min), C3 (60 min) and C7 (30 min) to SK-RC-52 cells by radioimmunoassay at 2 MBq/µg specific activity. (**c**) Radioimmunoactivity in HSA 20%, HS and PBS at 2 MBq/µg specific activity using the conjugate C3. (**d**) Radioimmunoactivity of [^177^Lu]DOTA(SCN)-cG250 at different activity concentrations using the conjugate C3. Blocking studies were performed with 50 µg of native cG250.

**Figure 6 pharmaceuticals-11-00132-f006:**
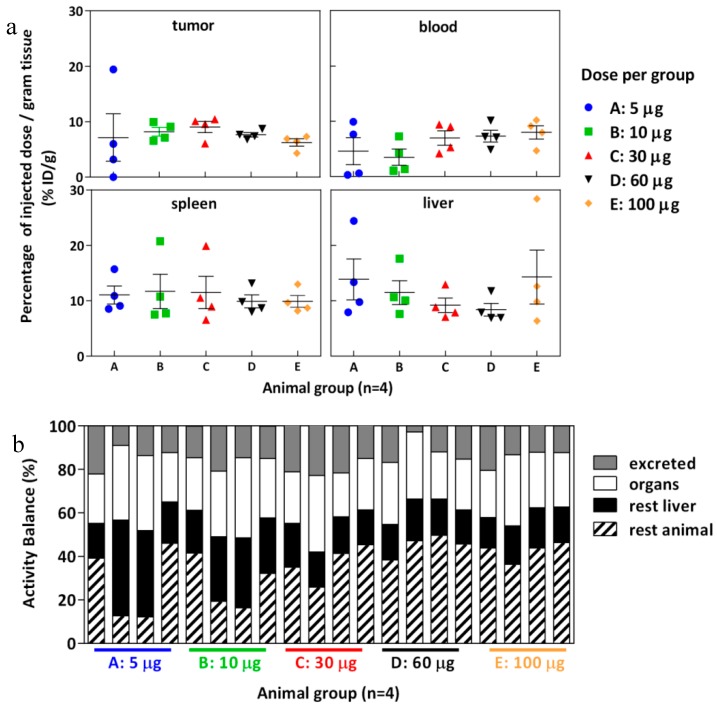
Biodistribution of [^177^Lu]DOTA(SCN)-cG250 from conjugate C7 at 48 h using total protein dose adjusted between 5 and 100 µg and 12 MBq per animal (*n* = 4). (**a**) Organs: tumor, blood, spleen and liver. (**b**) Activity balance.

**Figure 7 pharmaceuticals-11-00132-f007:**
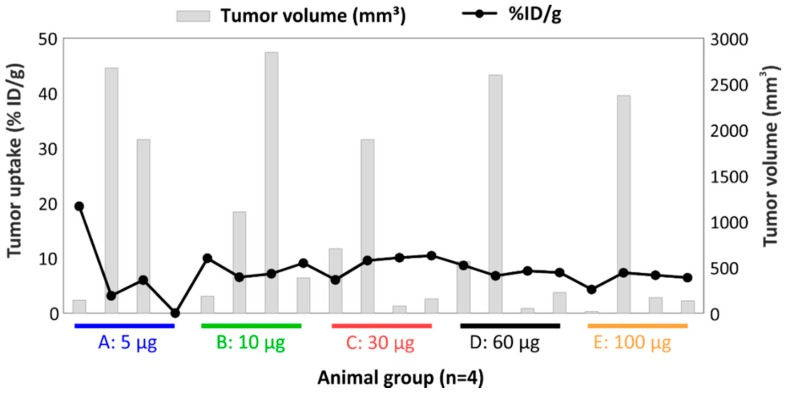
Relationship between tumor uptake and tumor volume per animal of the biodistribution of [^177^Lu]DOTA(SCN)-cG250 from conjugate C7 at 48 h using total protein dose adjusted between 5 and 100 µg and 12 MBq per animal (*n* = 4).

**Figure 8 pharmaceuticals-11-00132-f008:**
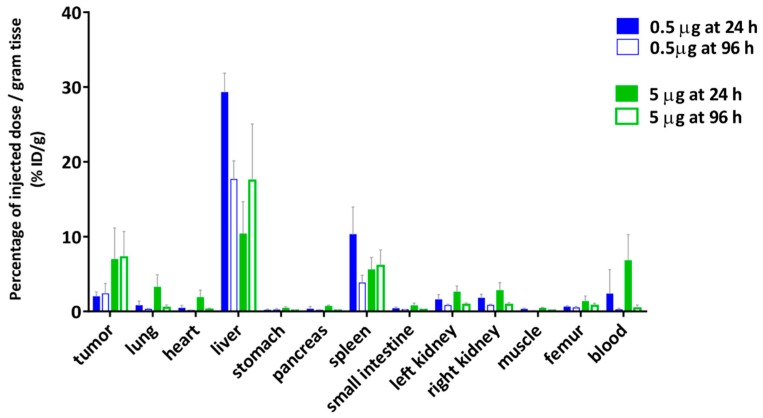
Biodistribution of [^177^Lu]DOTA(SCN)-cG250 from conjugate C7 using a 0.5 µg (2 MBq) and a 5 µg (18 MBq) protein dose per animal (*n* = 4).

**Figure 9 pharmaceuticals-11-00132-f009:**
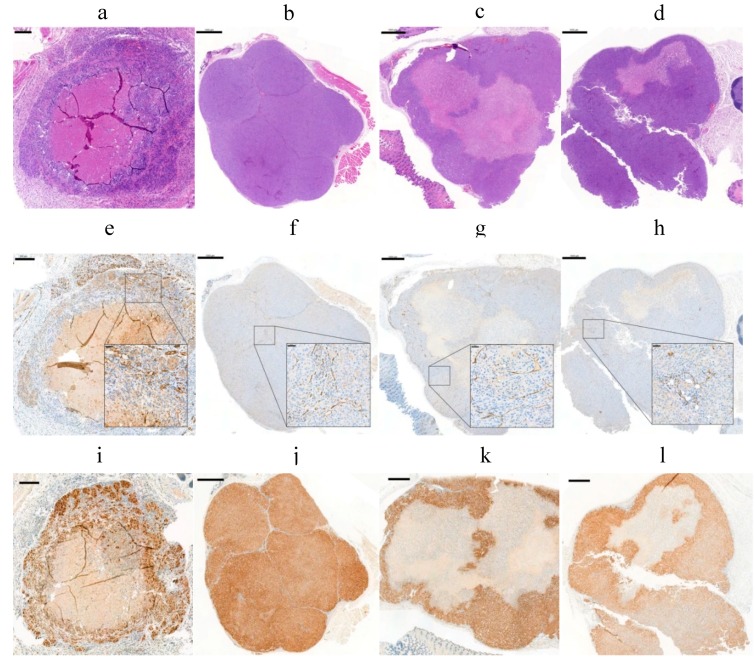
Evaluation of vascular density and hypoxia in FFPE from four different SK-RC-52 tumors: (**a**–**d**) (H&E) staining; (**e**–**h**) CD31 immunostaining; and (**i**–**l**) HIF1α staining. (**a**) 200 µm scale bar, 5x objective. (**b**–**d**) 1000 µm scale bar, 1.5x objective; (inset) 50 µm scale bar, 20x objective.

**Table 1 pharmaceuticals-11-00132-t001:** Average of DOTA(SCN) molecules per molecule of cG250 by mass spectrometry.

Conjugate	MW (Da)	Ratio Non Reduced	Ratio Reduced	
			HC	LC
C2 (90 min)	-	-	12–23 ^1^	6–7 ^1^
C3 (60 min)	154,187.07	8–10	3–4	1–2
C7 (30 min)	151,543.63	5–6	1–2	1–2

^1^ Values obtained by intact mass.

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
