# Peer review of "Evaluation of Radiolabeled Girentuximab In Vitro and In Vivo"

_pharmaceuticals, 2018, doi:10.3390/ph11040132_

Round 1
Reviewer 1 Report
I think you should include a list of abbreviation:
- It would make the article easier for us who aren’t hardcore biochemist to read.
- It could also help the you to remember to give all the abbreviations needed. I believe the you forgot to tell us that BFC probably is the abbreviations for “bifunctional chelator” and what about IHC, IC50, SE (SE chromatograms.), i.v. and PK ?
- In L104; you defined “HAS”, for the second time, see L 66. The same is done for cG250, CAIX in L105, but this may be OK since the first time was in the abstract
- On several occasions the you use girentuximab and cG250 in the same sentence, please, make up your mind if we want to use the abbreviation or not
- L42-44; You say: “targeted internal radiotherapy is currently being increasingly applied in 43 the treatment of a growing number of malignant diseases” [1-6]. But ref 1-6 are quite old (2003-2014) do you have some more recent ones?
Figure 5; Number of repeat, is not given
L. 400; Which type of animal is used?
L549; “in this range”, should it not be “above 60µg” ?
L587; please give the volume of 1M buffer used
L600: Is it correct that it the spectra was evaluated by chromeleon? Is it not you who evaluated the spectra and you use the Dionex software to do so?
Author Response
Thank you for consideration of our manuscript ‘Evaluation of radiolabeled Girentuximab in vitro and in vivo’ for publication. We appreciated the highly valuable comments of the reviewers and have revised the manuscript. The changes are highlighted in the revised manuscript and a point by point reply to the reviewer’s comments is listed below.
Reviewer 1
I think you should include a list of abbreviation:
- It would make the article easier for us who aren’t hardcore biochemist to read.
- It could also help the you to remember to give all the abbreviations needed. I believe the you forgot to tell us that BFC probably is the abbreviations for “bifunctional chelator” and what about IHC, IC50, SE (SE chromatograms.), i.v. and PK ?
- In L104; you defined “HAS”, for the second time, see L 66. The same is done for cG250, CAIX in L105, but this may be OK since the first time was in the abstract
- On several occasions the you use girentuximab and cG250 in the same sentence, please, make up your mind if we want to use the abbreviation or not
We have included a full list of abbreviation as Supplementary Table 2. Now cG250 is used throughout the paper consistently.
- L42-44; You say: “targeted internal radiotherapy is currently being increasingly applied in 43 the treatment of a growing number of malignant diseases” [1-6]. But ref 1-6 are quite old (2003-2014) do you have some more recent ones?
The citations reflect basic work that has been done on internal radiotherapy by means of monoclonal antibodies as investigated in our paper. In contrast to newer approaches using small molecules as carrier molecules or applying click-chemistry this work is not very recent. We have included a sentence referring also to the imaging potential of monoclonal antibodies and cited a more recent article investigating radiolabeled antibodies for prostate cancer imaging and dosimetry.
Figure 5; Number of repeat, is not given
N=3 is now mentioned
L. 400; Which type of animal is used?
Balb/c nu/nu nude mice were used throughout the study
L549; “in this range”, should it not be “above 60µg” ?
The wording has been changed
L587; please give the volume of 1M buffer used
The volume (50µl) was added
L600: Is it correct that it the spectra was evaluated by chromeleon? Is it not you who evaluated the spectra and you use the Dionex software to do so?
Yes, for the HPLC of the conjugation we used the Chromeleon Software.
Reviewer 2 Report
This manuscript describes a detailed evaluation of the parameters affecting conjugation reaction of p-SCN-Bn-DOTA to girentuximab. The authors also evaluated the factors affecting in vivo pharmacokinetics of the 177Lu-labeled antibody.
The conjugation reaction between an antibody and a chelating agent is governed by several parameters such as the concentrations of an antibody and a chelator (and the molar ratios between the two), reaction pH, reaction temperature, and reaction time, the adding rate of a chelator to an antibody, and so on. Unfortunately, the authors did not describe these parameters in the manuscript. In addition, the conjugation reaction presented in this manuscript was conducted under unusual conditions (too many chelators were used for the reaction). Usually, both the parental antibody and its chelate-conjugate show identical retention time on a size-exclusion HPLC analysis.
It is obvious from organic chemistry that isothiocyanates react with amine epsilon residues of proteins, due to their abundance (90 epsilon amine residues per IgG). The reduction of immunoreactivity by the heavy modification of an antibody with a chelating agent has also been well recognized.
The effect of mass amount of antibody on in vivo distribution including tumor accumulation has also been evaluated so far with controversial results. The relationship between the mass amount of antibody and in vivo tumor accumulation significantly depends on the antibody-xenograft models and presence or absence of circulating antigens.
Thus, the results of this manuscript are similar to those presented previously. However, if the 177Lu-DOTA-labeled cG250 is under consideration for clinical studies, the results in this study would provide a good basis for their future applications. In this case, the manuscript may be accepted for publication when the unusual conjugation reaction and subsequent characterization are revised.
Author Response
Thank you for consideration of our manuscript ‘Evaluation of radiolabeled Girentuximab in vitro and in vivo’ for publication. We appreciated the highly valuable comments of the reviewers and have revised the manuscript. The changes are highlighted in the revised manuscript and a point by point reply to the reviewer’s comments is listed below.
This manuscript describes a detailed evaluation of the parameters affecting conjugation reaction of p-SCN-Bn-DOTA to girentuximab. The authors also evaluated the factors affecting in vivo pharmacokinetics of the 177Lu-labeled antibody.
The conjugation reaction between an antibody and a chelating agent is governed by several parameters such as the concentrations of an antibody and a chelator (and the molar ratios between the two), reaction pH, reaction temperature, and reaction time, the adding rate of a chelator to an antibody, and so on. Unfortunately, the authors did not describe these parameters in the manuscript. In addition, the conjugation reaction presented in this manuscript was conducted under unusual conditions (too many chelators were used for the reaction). Usually, both the parental antibody and its chelate-conjugate show identical retention time on a size-exclusion HPLC analysis.
The reaction conditions were based on experience and literature reports. The conditions were described in details enabling their reproduction. In particular, the used ratio of chelators to antibody in the reaction is necessary to vary the final number of chelators per antibody just by changes in incubation time.
It is obvious from organic chemistry that isothiocyanates react with amine epsilon residues of proteins, due to their abundance (90 epsilon amine residues per IgG). The reduction of immunoreactivity by the heavy modification of an antibody with a chelating agent has also been well recognized.
The effect of mass amount of antibody on in vivo distribution including tumor accumulation has also been evaluated so far with controversial results. The relationship between the mass amount of antibody and in vivo tumor accumulation significantly depends on the antibody-xenograft models and presence or absence of circulating antigens.
We agree and think we have also cited previous work in a balanced manner. We also think the investigation of these parameters for this antibody is important for further development of targeting radioisotopes in general to CAIX.
Thus, the results of this manuscript are similar to those presented previously. However, if the 177Lu-DOTA-labeled cG250 is under consideration for clinical studies, the results in this study would provide a good basis for their future applications. In this case, the manuscript may be accepted for publication when the unusual conjugation reaction and subsequent characterization are revised.
One of the important findings of our study is the impact of the heterogeneous CAIX expression on tumor uptake. Therefore, we think clinical investigation should focus on small metastasis and eventually on single cells or cell-clusters. Therefore, we plan also the investigation of the short range alpha emitter Actinium-225 which might display advantages for treating disseminated disease. Concerning the conjugation procedure we did not consider it an unusual procedure. Nevertheless, we have expanded typical characterization of conjugated molecules with mass spectrometry methods.
Reviewer 3 Report
This manuscript entitles “Evaluation of radiolabeled Girentuximab in vitro and in vivo”. I would like to recommend this article for publication (major revision) after addition of the new data requested in revision.
1. This is thoroughly written manuscript describing in vitro and in vivo evaluation of [177Lu] cG250 in CAIX positive (SK-RC-52) and control (SK-RC-18) cell lines. The presentation is sound and the conclusions are justified based on the results.
2. Due to current limitations of clinical RIT, authors put much efforts in evaluating variables that might affect immunoreactivity of cG250 and would eventually help in clinically relevant RIT of RCCs.
3. Authors evaluated parameters affecting conjugation of cG250 with BFCs, numbers of BFCs/cG250, loss of immunoreactivity after conjugation, in vitro binding, in vivo pharmacokinetics, heterogeneity of CAIX expression in tumors etc.
4. There is already reported study of [177Lu] cG250 (phase-1, phase-2), [111In] cG250, [89Zr] cG250, [124I] cG250, [125I] cG250, [125I] cG250-IRDye800CW etc. with much more effective in vitro and in vivo data.
5. Manuscript should re-evaluate for grammatically errors.
6. Biodistribution (Figure 8) need additional data with blocking of CAIX (positive) and CAIX negative cell lines.
7. Please provide SPECT images of mice bearing CAIX positive (SK-RC-52) tumor, blocking and control (SK-RC-18) tumor to further confirm specificity and validation of your data in revised manuscript.
Author Response
Thank you for consideration of our manuscript ‘Evaluation of radiolabeled Girentuximab in vitro and in vivo’ for publication. We appreciated the highly valuable comments of the reviewers and have revised the manuscript. The changes are highlighted in the revised manuscript and a point by point reply to the reviewer’s comments is listed below.
This manuscript entitles “Evaluation of radiolabeled Girentuximab in vitro and in vivo”. I would like to recommend this article for publication (major revision) after addition of the new data requested in revision.
1. This is thoroughly written manuscript describing in vitro and in vivo evaluation of [177Lu] cG250 in CAIX positive (SK-RC-52) and control (SK-RC-18) cell lines. The presentation is sound and the conclusions are justified based on the results.
2. Due to current limitations of clinical RIT, authors put much efforts in evaluating variables that might affect immunoreactivity of cG250 and would eventually help in clinically relevant RIT of RCCs.
3. Authors evaluated parameters affecting conjugation of cG250 with BFCs, numbers of BFCs/cG250, loss of immunoreactivity after conjugation, in vitro binding, in vivo pharmacokinetics, heterogeneity of CAIX expression in tumors etc.
4. There is already reported study of [177Lu] cG250 (phase-1, phase-2), [111In] cG250, [89Zr] cG250, [124I] cG250, [125I] cG250, [125I] cG250-IRDye800CW etc. with much more effective in vitro and in vivo data.
5. Manuscript should re-evaluate for grammatically errors.
The manuscript has been checked for language again and several incorrect expressions have been replaced or improved.
6. Biodistribution (Figure 8) need additional data with blocking of CAIX (positive) and CAIX negative cell lines.
An additional experiment showing the ability to block the tumor accumulation by unlabeled antibodies was described and shown in the supplemental material.
7. Please provide SPECT images of mice bearing CAIX positive (SK-RC-52) tumor, blocking and control (SK-RC-18) tumor to further confirm specificity and validation of your data in revised manuscript.
Unfortunately a small animal SPECT is not available to us. Therefore, this interesting suggestion is beyond the scope of our ability.
Round 2
Reviewer 2 Report
I am sorry to say that the present conjugation procedure by the authors is quite far from the standard procedure. The amount of the bifunctional chelator and the chelator-to-antobody molar ratios were too high. Could the authors prepare clear solution of antibody at the concentration of 100 mg/mL? The conjugation procedure referred by the authors came from a patent, indicating that the procedure is not authorized by experts of this field.
Reviewer 3 Report
This manuscript entitles “Evaluation of radiolabeled Girentuximab in vitro and in vivo”. I would recommended the article for publication as previous comments are followed by Authors and incorporated in the revised manuscript accordingly.